# Enhancement of the Protective Activity of Vanillic Acid against Tetrachloro-Carbon (CCl_4_) Hepatotoxicity in Male Rats by the Synthesis of Silver Nanoparticles (AgNPs)

**DOI:** 10.3390/molecules27238308

**Published:** 2022-11-28

**Authors:** Eman S. Alamri, Haddad A. El Rabey, Othman R. Alzahrani, Fahad M. Almutairi, Eman S. Attia, Hala M. Bayomy, Renad A. Albalwi, Samar M. Rezk

**Affiliations:** 1Department of Nutrition and Food Science, University of Tabuk, Tabuk 47512, Saudi Arabia; 2Biochemistry Department, Faculty of Science, University of Tabuk, Tabuk 47512, Saudi Arabia; 3Bioinformatics Department, Genetic Engineering and Biotechnology Research Institute, University of Sadat City, Sadat City 32897, Egypt; 4Department of Biology, University of Tabuk, Tabuk 47512, Saudi Arabia; 5National Nutrition Institute, Ministry of Health, Cairo 4262114, Egypt; 6Department of Food Science and Technology, Damanhour University, Damanhour 22511, Egypt; 7Clinical Nutrition Department, Mahalla Hepatology Teaching Hospital, El-Mahalla El-Kubra 4260010, Egypt

**Keywords:** vanillic acid, silver nanoparticles, silymarin, CCl_4_, hepatotoxicity

## Abstract

In the current study, the hepatoprotective activity of vanillic acid, silymarin, and vanillic acid-loaded silver nanoparticles (AgNPs) against CCl_4_-induced hepatotoxicity was tested in male rats for four weeks. Thirty male rats were divided into five groups (*n* = 6). The 1st group was a negative control, the 2nd group was a positive control, the 3rd group was treated with 100 mg/kg b.w. of vanillic acid, the 4th group was treated with 100 mg/kg b.w. of vanillic acid–AgNPs, and the 5th group was treated with 50 mg/kg b.w. of silymarin. The CCl_4_-induced hepatic toxicity in the 2nd group was revealed by the liver function and all other biochemical tests. Liver enzymes, bilirubin, lipid peroxidation, lactate dehydrogenase, and interleukin-6 were elevated, whereas, total protein, antioxidant enzymes, and irisin were decreased compared to the negative control. The hepatic tissues were also injured as a result of the CCl_4_-induced hepatotoxicity. Treating the hepatotoxic rats with vanillic acid moderately protected the rats of the 3rd group, whereas treatment with vanillic AgNPs and silymarin in G4 and G5, respectively, greatly protected the rats against the CCl_4_ hepatotoxicity, approaching the normal biochemical levels and liver tissue appearance. The biochemical tests were confirmed by the histological investigations of liver tissue.

## 1. Introduction

The liver is the largest organ in the human body that plays a large role in the removal and disposal of toxic substances, but it may be damaged as a result of excessive doses of certain drugs or chemical compounds [1,2,3]. Liver diseases have many other causes such as excessive alcohol drinking, autoimmune disorder, or infection leading to the production of free radicals through numerous biochemical pathways that damage proteins, DNA, and lipids [4,5]. Carbon tetrachloride (CCl_4_)—which is a hepatotoxin—is used for experimental purposes to induce liver toxicity in animal models [2,5,6].

Antioxidants such as phenolic compounds play a major role in alleviating liver toxicity [2,5,7]. The phenolic derivative, vanillic acid is found in vanilla and in lower quantities in other species such as female ginseng, benzoin, doum, soybeans, olives, oranges, guava, etc., and is used in cosmetics, fruit flavorings, cigarettes, pharmaceuticals, alcoholic drinks, flavoring agents, and fragrances [8,9].

Recent research shows that vanillic acid has many pharmacological activities due to its antioxidant and free radical scavenging activity by fighting oxidative stress via tissue activation [10,11,12]. It was described in neuropharmacological studies as a sedative which delays sleeping, increasing the length of sleeping time, and decreasing the locomotor activity and exploratory behavior in mice [13]. It also has an antidepressant effect through decreasing neuroplasticity impairment and battling depression [14]. It has also acted as a dose-dependent suppressive to the writhing reaction induced by acetic acid [15]. In addition, vanillic acid also showed antihypertensive activity for a persistent rise in blood pressure and it has decreased plasma nitrite/nitrate concentrations [11,16]. Furthermore, vanillic acid administration also decreased plasma levels of interleukin (IL)-6 in a dextran sodium sulfate (DSS)-treated group of the colitis model [17]. In addition, vanillic acid also showed an anticancer activity through inhibiting tumor development [18]. Vanillic acid also showed hepatoprotective activity in chronic liver injury by minimizing hepatic fibrosis [19]. 

Vanillic acid was also used in wound healing by suppressing microphthalmia-associated transcription factor (MITF) and melanogenic enzyme sinB16F10 cell expression, and decreasing melanin levels and tyrosinase activity with or without melanocyte-stimulating hormone (MSH) stimulation [20]. Vanillic acid also has an antifungal and antiradical activity against specific fungal strains [21]. In addition, vanillic acid was characterized as a novel neuroprotective agent due to the antioxidant activity of the methoxy group, which is characterized by radical scavenging activity [9,10,22]. It also has genotoxic and antigenotoxic, as well as potential cytotoxic, effects [23]. It treated the injury of the acute myocardial hypoxia/reoxygenation by decreasing the oxidative stress of H9c2 cells [24]. It also controlled mice neurotoxicity, which was induced by lipopolysaccharide by regulating the c-Jun N-terminal kinase in the brain [25]. It also inhibited inflammatory pain through nuclear factor Kappa B (NF-κB) activation, and neutrophil recruitment, oxidative stress, and cytokine production inhibition [26].

Applications of nanoparticle technology in the field of nanomedicine via nano-delivery systems has increased during the last decade where these nanoscale range materials are applied as diagnostic tools or delivery agents of therapeutic materials to the targeted sites under a precise scientific discipline [27,28,29,30]. Nanotechnology has helped in treating multiple chronic human diseases through the site-specific and target-oriented delivery of specific medicines [31,32]. The biomedical characteristics and uses of silver-based nanostructures were extensively characterized by Burdușel et al. [33]. Silver nanoparticles succeeded in combating extensively drug-resistant *Pseudomonas aeruginosa* [34], and inhibiting vancomycin resistance in *Staphylococcus aureus* [30].

In addition, silymarin (extracted from an herb known as *Silybum marianum*) is widely used in herbal and complementary medicine due to its anti-inflammatory and antioxidant activity in protecting against liver toxicity [2,5].

The objective of the current study is to explore the protective effect of vanillic acid, vanillic acid-loaded silver nanoparticles (AgNPs), and silymarin against CCl_4_-induced hepatotoxicity in male rats. 

## 2. Results and Discussion

To ensure the synthesis of vanillic acid nanoparticles, an emergence of dark brown color in the vanillic acid solution following the addition of silver nitrate was formed indicating silver nanoparticle synthesis, while the solution stayed white without the addition of silver nitrate as shown in Appendix A [29,35].

The synthesis of the reduced AgNPs of vanillic acid in the solution was visualized under UV-Vis spectrum analysis. The UV-Vis spectrum showed that the synthesized metal nanoparticles exhibited surface plasmon resonance and their characteristic surface plasmon resonance band occurred at 450 nm as seen in Figure 1 [27,36]. The transmission electron microscope (TEM) image of the synthesized silver nanoparticles of vanillic acid is shown in Figure 2. The AgNPs were spherical in shape and their size ranged from 11.0–14.1 nm [29,37]. The AgNPs’ structure is based on the electron plasmon oscillations of their free surface seen by the UV-Vis spectra, whereas the shape and size of their peaks in the spectra are ascribed to the surface plasmon resonance (SPR) peaks, and the presence of peaks at 450 nm indicates that there was no significant difference in the AgNPs’ size [35,37]. In addition, the DLS spectrum of the synthesized silver nanoparticles gave an average particle size of 15.3 nm as shown in Figure 3 [38].

### 2.1. Effect of CCl_4_-Induced Liver Toxicity on Liver Enzymes

The CCl_4_-induced toxicity elevated liver enzymes (ALT, AST, ALP, and GGT) in the positive control group (G2) compared to the negative control group (G1) as shown in Figure 4A–D, respectively, and the Appendix A. The CCl_4_-induced hepatotoxicity raised the free radical content and the oxidative stress, and consequently damaged the live cells and released the liver function enzymes (ALT, AST, ALP, and GGT) in the serum, which increased their levels in the serum [2,5,6]. Treating the hepatotoxic rats with vanillic acid, vanillic AgNPs, and silymarin in G3, G4, and G5, respectively, greatly improved the altered liver enzymes and nearly restored them to the negative control group levels. The beneficial effects of vanillic acid and its AgNPs is attributed to its antioxidant activity acquired by the radical scavenging activity of its methoxy group [10,22,39], whereas silymarin is one of the best-known drugs in protecting the liver [22,40]. 

Treatment with vanillic acid-loaded silver nanoparticles and silymarin in G4 and G5, respectively, improved the hepatic enzymes more than vanillic acid in G3. The loading of vanillic acid on the silver nitrate nanoparticles greatly increased its delivery reflected by the returning of almost all the altered liver enzymes [22,41].

### 2.2. The Effect of CCl_4_ on Total Protein and Bilirubin

In G2, CCl_4_-induced toxicity elevated bilirubin and direct bilirubin, whereas it decreased total protein and albumin compared to the negative control group (G1) as shown in Table 1. This is due to the higher oxidative stress that affected the liver [39]. Hepatotoxicity caused a disturbance in metabolism, reduced protein synthesis, and increased bilirubin in the regular positive control group [40]. The levels of total protein (TP) were reduced in CCl_4_ rats indicating lysis in the number of hepatocytes, which led to a reduced ability of the liver to synthesize proteins and an increase in bilirubin [2,41]. However, treating the hepatotoxic rats with vanillic acid, vanillic acid-loaded silver nanoparticles, and silymarin in G3, G4, and G5, respectively, greatly increased total proteins and decreased bilirubin and direct bilirubin approaching the negative control group levels. This alleviating effect is ascribed to the antioxidant activity of vanillic acid and the treatment effect of silymarin [10,22]. 

Treatment with vanillic acid-loaded silver nanoparticles and silymarin in G4 and G5, respectively, improved the altered protein and bilirubin levels more than vanillic acid in G3. The synthesized vanillic acid–AgNPs improved the drug delivery process and as a result, enhanced the altered protein and bilirubin levels to approaching that of the negative control group [35,36,37]. The protective activity of silymarin also assisted in restoring the TP and bilirubin levels [27,42].

### 2.3. The Effect of CCl_4_ on Lactate Dehydrogenase (LDH), Interleukin-6 (IL-6), and Irisin 

The LDH and IL-6 levels were increased in the positive control group (G2), whereas irisin was decreased as a result of the CCl_4_-induced toxicity as shown in Figure 5A–C and the the Appendix A. The increase in LDH and IL-6 and the decrease in irisin occurred as a result of the increase in the production of reactive metabolites and oxidative stress, and thus eventually led to an increase in toxicity resulting from CCl_4_, because CCl_4_ toxicity increases pro-inflammatory cytokines, such as IL-6, that are released at the site of intensive liver damage and produce much more reactive oxygen species (ROS) that induce oxidative stress [2,6,39]. 

However, treating the CCl_4_-induced toxic rats in G3, G4, and G5 with vanillic acid, AgNPs-loaded vanillic acid, and silymarin, respectively, reduced the oxidative stress and normalized the redox system and the pro-inflammatory cytokines and cytokines (IL-6) [2,6]. The beneficial effects of vanillic acid and its AgNPs is attributed to its antioxidant activity acquired by the radical scavenging activity of its methoxy group [10,22,39], whereas silymarin is a known drug for protecting the liver [22,41].

In addition, the effect of vanillic acid AgNPs protected the liver more than the normal vanillic acid due to the enhanced drug delivery mechanism related to the synthesis of AgNPs and it greatly increased irisin and decreased LDH and IL-6 restoring them nearly to the negative control group levels [28,35,37].

### 2.4. The Effect of CCl_4_-Induced Hepatotoxicity on Antioxidant Enzymes and Total Antioxidant Capacity

The CCl4-induced hepatotoxicity decreased the total antioxidant capacity (TAC) and all antioxidant enzymes (Catalase, SOD, GST, and GSH) in the liver tissue homogenate of the positive control group (G2) compared to the negative control group (G1), as shown in Table 2. This decrease in levels of all antioxidant enzymes and the TAC raised the oxidative stress that affected the liver causing hepatotoxicity [2,6,39].

Treating the hepatotoxic rats with vanillic acid, vanillic acid-loaded silver nanoparticles, and silymarin in G3, G4, and G5, respectively, greatly increased the levels of the altered studied antioxidants (GSH, GST, CAT, and SOD) and the total antioxidant capacity and nearly restored them to the negative control group levels [5,6,7]. 

The results also showed that the treatment with synthesized AgNPs of vanillic acid caused the reduction of oxidative stress in G4 more than that of vanillic acid in G3 due to the enhanced drug delivery mechanism of vanillic acid–AgNPs over vanillic acid in G3 [27,28], while silymarin treatment increased the antioxidant levels in the liver tissue homogenate due to the protection of the liver that was reflected by restoring the studied antioxidants and TAC nearly to their normal levels as in the negative control (G1) [41,42,43]. Although an increase in ROS is very harmful, a certain level of ROS is important for adjusting the redox cell signals to maintain cellular homeostasis [44].

### 2.5. The Effect of CCl_4_-Induced Hepatotoxicity on Lipid Peroxidation as Revealed by Malondialdehyde (MDA)

The CCl_4_-induced hepatotoxicity increased lipid peroxidation as revealed by malondialdehyde (MDA) in the liver tissue homogenate of the positive control group (G2) compared to the negative control group (G1) as shown in Figure 6 and Appendix A. This increase in the level of lipid peroxidation occurred because the CCl_4_-induced hepatotoxicity increased liver inflammation and the production of free radicals that increased oxidative stress and lipid peroxidation as indicated by the increased malondialdehyde (MDA) [2,6,39]. In addition, the increase in the level of lipid peroxidation was followed by an effect on the nucleic acids and cellular proteins that ultimately reduced the function of hepatocytes [2,6].

On the other hand, treating the hepatotoxic rats with vanillic acid, vanillic acid-loaded AgNPs, and silymarin in G3, G4, and G5, respectively, greatly decreased lipid peroxidation in the liver tissue homogenate and nearly restored the MDA levels to those of the negative controls. This ensured the antioxidant activity of the vanillic acid, that is acquired by the scavenging activity of its methoxy group, and consequently reduced the oxidative stress [7,10,22,39]. Moreover, treatment with vanillic acid–AgNPs in G4 showed a more protective effect, as revealed by the decrease in MDA and the antioxidant redox system, than did the vanillic acid in G3, due to the enhanced drug delivery mechanism of the AgNPs [27,28]. Silymarin in G5 showed greater protection of the liver tissues and consequently decreased the lipid peroxidation [41,42,43].

### 2.6. The Effect of CCl_4_-Induced Hepatotoxicity on Liver Histopathology

The liver tissues of the negative control group (G1) are shown in Figure 7A. The image shows the preserved normal architecture without inflammation or necrosis or any abnormalities. The CCl_4_-induced toxicity in the positive control group (G2) showed drastic changes in liver tissues as shown in Figure 7B. The hepatic tissues showed distorted hepatic plates, hepatocytes ballooning with mild fatty changes, and dense portal tract mononuclear infiltrate that may be caused by the higher oxidative stress that occurred as a result of the CCl_4_-induced toxicity [2,5,6]. 

Treating CCl_4_-induced toxicity with vanillic acid in G3 partially improved the liver tissues as shown in Figure 7C. It showed mild inflammation of hepatocytes and mild fatty changes. This improvement is ascribed to the antioxidant activity of vanillic acid. On the other hand, treating the CCl_4_ hepatotoxicity in G4 and G5 with vanillic acid-loaded AgNO_3_ nanoparticles and silymarin, respectively, showed nearly normal hepatic tissues with normal hepatic architecture (Figure 7D). This ameliorative effect occurred because the AgNPs of vanillic acid in G4 enhanced the drug delivery mechanism and reduced the oxidative stress more than the vanillic acid in G3 [10,22,39]. Similarly, silymarin also protected the liver by reducing oxidative stress within the liver tissues and nearly restored them to their normal architecture as shown in Figure 7E [2,41,42,43].

In this study, silymarin showed a high protective activity against the CCl_4_-induced hepatic toxicity because of its antioxidant, cytoprotection, radical scavenging, and anti-inflammatory properties [2,5] by the blocking and adjustment of cell transporters, p-glycoprotein, and estrogenic and nuclear receptors, and the reduction of TNF-α, as well as the protective effects on erythrocyte lysis and cisplatin-induced acute nephrotoxicity [45].

## 3. Materials and Methods

### 3.1. Chemicals and Reagents 

Vanillic acid and other chemicals used in this study were from Sigma (St. Louis, MO, USA), unless nominated to other sources. Virgin olive oil was obtained from the local market, whereas silymarin was obtained from SEDICO Company, Giza, Egypt.

### 3.2. Synthesis of Silver Nanoparticles from Vanillic Acid

The silver nitrate nanoparticles of vanillic acid were synthesized by the photo-mediated method for reducing silver ions as follows: two grams of vanillic acid was dissolved in 10 mL of ethanol and then completed by distilled water up to 100 mL; twenty milligrams of silver nitrate crystals (Sigma-Aldrich, St. Louis, MO, USA) was added to the vanillic acid solution and thoroughly mixed by a magnetic stirrer; the pH was adjusted by 1.0 M NaOH to 10.0. The color of the solution was initially whitish, then changed to yellow, and subsequently to dark brown ensuring the formation of the silver nanoparticle [30,35]. 

#### Characterization of Silver Nanoparticles

Ultraviolet visible spectroscopy

The AgNPs surface plasmon resonance was detected by measuring the UV-Vis spectra of the synthesized particles using a UVS-85 spectrophotometer (AccuLab, New York, NY, USA) spanning the range from 300 to 800 nm [36]. 

2.Transmission electron microscope (TEM)

The synthesized AgNPs size, morphology, form, purity, and assembly were characterized using a transmission electron microscope. For electron microscope investigations, samples were prepared by placing 2–5 μL droplets of the sample onto a sheet of parafilm, followed directly by creating EM grids on the sample. Subsequently, the samples were wiped using filter paper and the grids were inserted in the sample petri dish [37].

3.Dynamic light scattering (DLS)

The size and distribution of the synthesized AgNPs dispersed in a liquid sample of silver nanoparticles were estimated using the NICOMP Nano ZLS (Z3000 zls) particle sizing device (Entegris, Germany). The AgNPs’ size was calculated using dynamic light scattering (DLS). The samples were diluted ten times in deionized water before analysis and 250 μL of the suspension was transferred to a low-volume disposable cuvette. Prior to measuring, the sample was equilibrated at 20 °C for 2 min [30,38].

### 3.3. Test Animals and the Experiment Design

Thirty adult Sprague–Dawley male rats (weighing 175.25 ± 8.66 g) were supplied by the Agricultural Research Center, Giza, Egypt, and then housed in polypropylene cages (*n* = 6) under observation for two weeks before the experiment began under standard lab conditions (humidity = 60%, room temperature = 23 °C, and the light cycle = 12 h) according to the guidelines of the National Institutes of Health (NIH) for the care and use of laboratory animals (NIH Publication, Number 85–23, Revised 1985). This study was also ethically approved by The Institutional Research Board (IRB) Committee of the Faculty of Medicine, Mansoura University, Egypt (Approval code: 00-0178). Water and food were supplied *ad libitum* during the experiment. 

The thirty rats were divided into five groups (6 rats each) as follows: the 1st group (G1) was the negative control that received only 1 mL/kg b.w. of 1:1 olive oil in distilled water (at 8:00 am) via intraperitoneal injection (in the lower left or right quadrant of the abdomen) on the 1st and 4th day every week until the end of the experiment. Liver damage was induced in the other 24 rats by injecting rats on the 1st and 4th day every week (at 8:00 am) with CCl_4_ (1 mL/kg body weight) using the intraperitoneal injection with an equal amount of olive oil (1:1) for four weeks until the end of the experiment [6], and these rats were divided into four groups. The 2nd group (G2) was the positive CCl_4_ hepatotoxic control group and received the regular CCl_4_, only. The 3rd group (G3) was treated with 100 mg/kg b.w. of vanillic acid (1 mL of 1% vanillic acid/each 200 g rat) [11]. The 4th group (G4) was treated with 100 mg/kg b.w. of silver nitrate nano vanillic acid (1 m of 1% np vanillic acid/each 200 g rat). The 5th group (G5) was treated with 50 mg/kg b.w. of silymarin (as a positively treated group) [2]. Rats in all groups were treated with the intraperitoneal injection.

### 3.4. Blood Collection

By the end of the experiment, all rats were euthanized with 2.5 % halothane (1:20 halothane vapor: air ratio with a flow rate of 70% V/min), and the abdomen was dissected. The blood samples were collected from the heart, centrifuged for 5 min at 3000× *g* rpm, and the serum was transferred into sterilized tubes for biochemical analysis. In addition, the liver was removed from the abdomen, washed in cold saline, and then divided into two pieces; one piece was kept on ice for homogenate preparation and the other piece was fixed in 10% formalin for histopathological examination.

### 3.5. Liver Tissue Homogenate Preparation 

Homogenates of liver tissues were prepared as described in El Rabey et al. [2]. The ice-cold liver tissue was rinsed in ice-cold saline solution and homogenized at 4 °C in 0.1 M Tris–HCl buffer (pH 7.4) using a Teflon homogenizer (Thomas Scientific, Swedesboro, NJ, USA): the homogenate was then centrifuged to remove any debris at 13,000× *g*. The supernatant was kept in the fridge at 4 °C for estimating antioxidant enzymes and lipid peroxidation.

### 3.6. Biochemical Tests

Liver function enzyme activity was estimated in serum with kits from Sigma Aldrich Company (USA) according to the manufacturer’s instructions. The following enzyme activities were estimated: aspartate aminotransferase (AST) activities and alanine aminotransferase (ALT) were estimated according to the method of Reichling and Kaplan [46], whereas the activity of alkaline phosphatase was estimated according to the method of Mossner et al. [47]. In addition, gamma glutamyl transferase (GGT) was estimated using the Abcam kit (Abcam, Waltham, MA, USA) according to the instruction of the supplier.

The serum irisin level was determined using a commercial kit from Cono Biotech Co., Ltd. (Xi’an, China), whereas the interleukin-6 (IL-6) level was determined using a kit from MyBiosource (San Diego, CA, USA). All analyses were done according to the instructions of the suppliers.

Superoxide dismutase (SOD), catalase (CAT), and glutathione-s-transferase (GST) activities and total antioxidant capacity (TAC) were estimated in the liver tissue homogenate using a Biodiagnostic Kit (Giza, Egypt). In addition, for lipid peroxidation estimation, the malondialdehyde (MDA) level in the liver tissue homogenate was estimated using a Biodiagnostic Kit (Egypt). All the analyses carried out in this study were according to the instructions of the kit suppliers. 

### 3.7. Histological Study

The 10% formalin-fixed liver tissue was dehydrated with an ethanol series, cleared in xylene, embedded in paraffin, and then 5 µ microtomic sections were prepared. Finally, the resulting sections were stained with hematoxylin and eosin (H & E) according to Bancroft and Stevens [48].

### 3.8. Statistical Analysis

All data were presented as mean ± standard error (SE) and then analyzed using the one-way analysis of variance (ANOVA) for comparison of means between groups. All analyses were done using the statistical package program (SPSS) version 17.0 [49]. 

## 4. Conclusions

CCl_4_-induced severe hepatotoxicity in male rats was investigated through altered biochemical and histological investigations. Treatment of the CCl_4_-induced hepatotoxicity with vanillic acid, vanillic acid AgNPs, and silymarin significantly protected the liver against the CCl_4_-induced hepatotoxicity as revealed by the improved biochemical and histological tests. In this study, we measured specific biochemical parameters, such as total proteins, bilirubin, irisin, LDH, IL-6, liver function enzymes, antioxidants, and lipid peroxidation to assess CCl_4_-induced hepatotoxicity and its mitigation after administration of vanillic acid, vanillic acid AgNPs, and silymarin. The synthesized vanillic acid–AgNPs were examined to ensure their proper synthesis. They were also photographed under TEM and characterized using specialized software to detect their shape, size, and distribution (the average size was 15.3 nm). These beneficial effects of vanillic acid and its AgNPs are attributed to its antioxidant activity acquired by the radical scavenging activity of its methoxy group that has the potential for improving LDH, irisin, IL-6, and other biomarkers, showing a restored liver function, and liver tissues were restored nearly to the normal state as in the negative control group. In addition, the AgNPs–vanillic acid in G4 was much more effective than vanillic acid in G3. Furthermore, treatment with silymarin in G5 showed the greatest protection to the liver against the CCl_4_-induced hepatotoxicity.

## Figures and Tables

**Figure 1 molecules-27-08308-f001:**
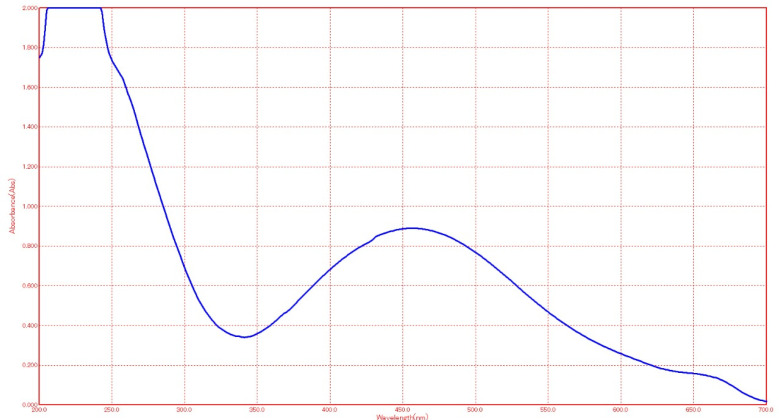
The silver surface plasmon resonance band as seen under UV-Visible spectra. The peak appears at 450 nm.

**Figure 2 molecules-27-08308-f002:**
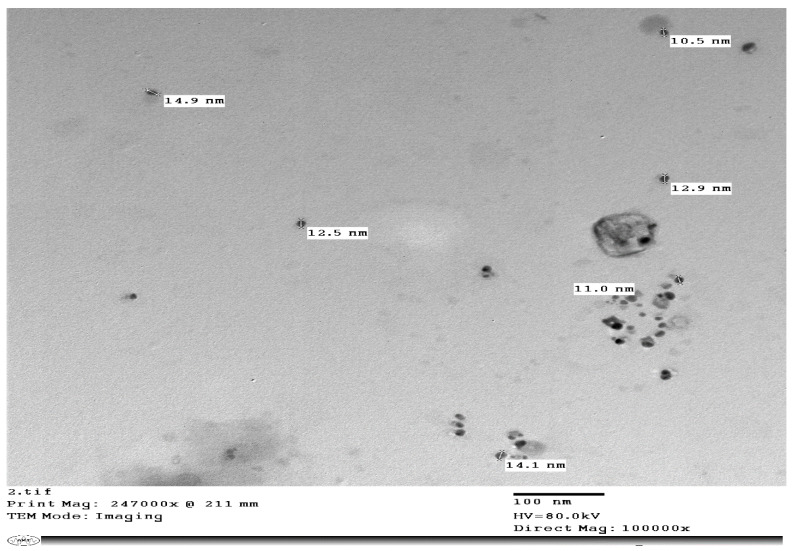
The AgNPs obtained from vanillic acid as seen by TEM. Particle size ranged from 11.0 nm to 14.1 nm.

**Figure 3 molecules-27-08308-f003:**
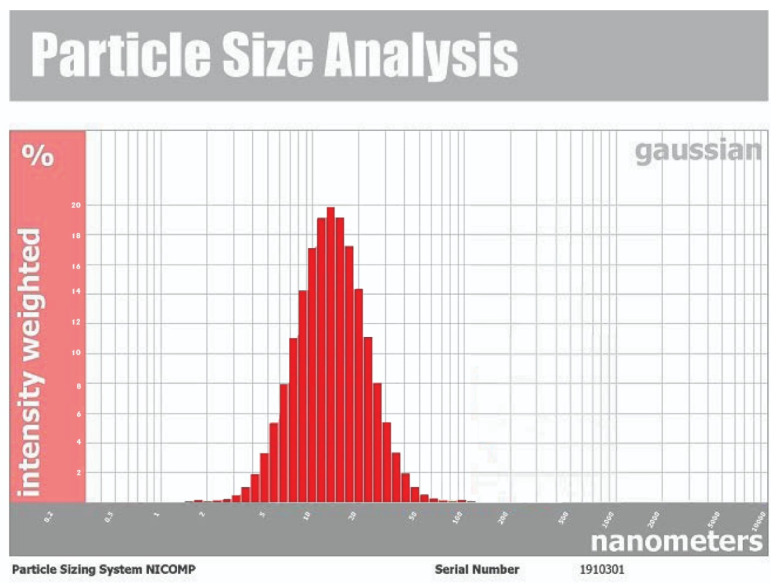
**The** DLS spectrum for silver nanoparticles. The average particle size was 15.3 nm.

**Figure 4 molecules-27-08308-f004:**
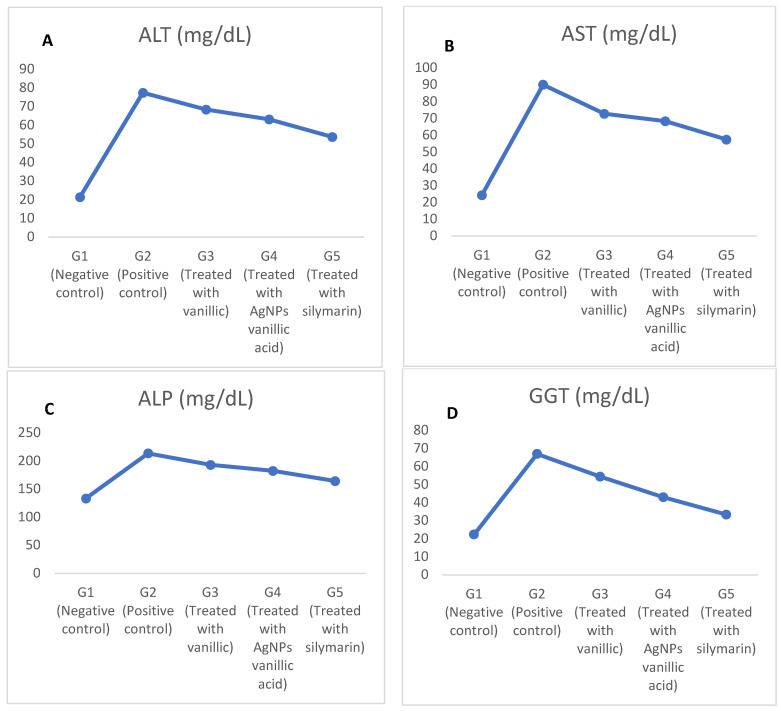
The effects of vanillic acid-loaded silver nanoparticles on serum liver enzymes in CCl_4_ liver toxicity-induced rats. (**A**): ALT, (**B**): AST, (**C**): ALP, and (**D**): GGT.

**Figure 5 molecules-27-08308-f005:**
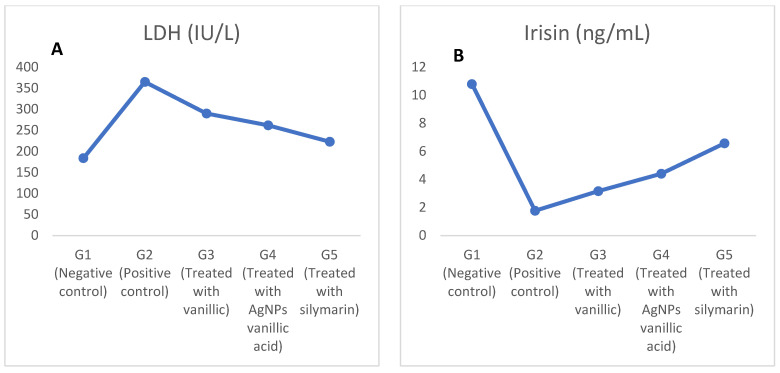
The effects of vanillic acid-loaded silver nanoparticles on serum lactate dehydrogenase (LDH), irisin, and IL-6 in CCl_4_ liver toxicity-induced rats. (**A**): LDH, (**B**): Irisin, and (**C**): IL-6.

**Figure 6 molecules-27-08308-f006:**
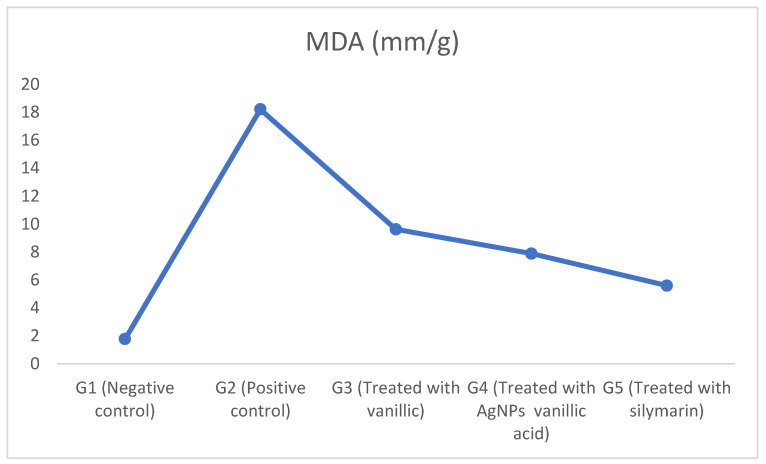
The effects of vanillic acid-loaded silver nanoparticles on malondialdehyde (MDA) in liver tissues of CCl_4_ liver toxicity-induced rats.

**Figure 7 molecules-27-08308-f007:**
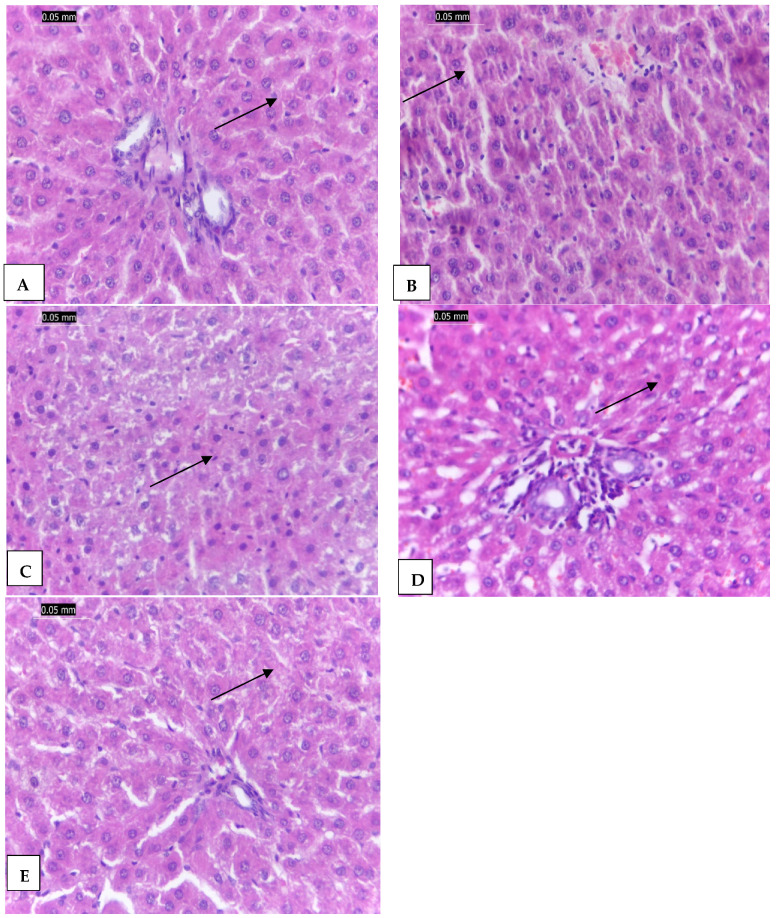
(**A**): The control negative group (G1) showing normal hepatic tissues architecture without any necrosis or inflammation (arrow), (**B**): The CCl_4_ treated group (G2) showing distorted hepatic plates, fatty hepatocytes (arrow), and dense portal tract mononuclear infiltrate, (**C**): G3 treated with vanillic acid showing mild inflammation (arrow), (**D**): The G4 group treated with vanillic AgNPs showing mild inflammation to nearly normal hepatic tissues (arrow), and (**E**): The silymarin treated group (G5) showing nearly normal hepatic tissues (arrow). (H & E × 400).

**Table 1 molecules-27-08308-t001:** The effects of vanillic acid-loaded silver nanoparticles on serum protein and bilirubin in CCl_4_ liver toxicity-induced rats.

Parameters mg/dL	Statistics	G1 (Negative Control)	G2 (Positive Control)	G3 (Treated with Vanillic)	G4 (Treated with AgNP–Vanillic Acid)	G5 (Treated with Silymarin)
Total Protein	Mean ± SE	6.53 ± 0.11 ^a^	3.34 ± 0.05 ^e^	4.84 ± 0.04 ^d^	5.58 ± 0.06 ^c^	6.07 ± 0.03 ^d^
	LSD0.05 = 0.259					
	*t*-test	-	1.16 ***	−1.80 ***	−1.86 ***	−0.48 ***
Albumin	Mean ± SE	4.23 ± 0.05 ^a^	1.43 ± 0.05 ^e^	2.30 ± 0.03 ^d^	2.80 ± 0.03 ^c^	3.56 ± 0.04 ^d^
	LSD0.05 = 0.143					
	*t*-test	-	0.56 ***	−0.55 ***	−0.82 ***	−0.21 ***
Total Bilirubin	Mean ± SE	0.43 ± 0.01 ^a^	1.82 ± 0.02 ^e^	1.14 ± 0.03 ^b^	0.86 ± 0.02 ^c^	0.66 ± 0.01 ^d^
	LSD0.05 = 0.065					
	*t*-test	-	−1.51 ***	1.97 ***	2.60 ***	1.04 ***
Direct Bilirubin	Mean ± SE	0.3 ± 0.00 ^a^	1.22 ± 0.80 ^e^	0.74 ± 0.50 ^b^	0.64 ± 0.40 ^c^	0.44 ± 0.30 ^d^
	LSD0.05 = 0.028					
	*t*-test	-	0.76 ***	−1.06 ***	−1.58 ***	−1.42 ***

*t*-test values, ***: means significant at *p* < 0.001. a, b, c, d, or e are ANOVA analysis significantly different at *p* < 0.05, LSD: least significant difference.

**Table 2 molecules-27-08308-t002:** The effects of vanillic acid-loaded silver nanoparticles on TAC and antioxidant enzymes in liver tissues of CCl_4_ liver toxicity-induced rats.

Parameters Mg Liver Tissue/dL	Statistics	G1 (Negative Control)	G2 (Positive Control)	G3 (Treated with Vanillic)	G4 (Treated with AgNPs–Vanillic Acid)	G5 (Treated with Silymarin)
GST U/g	Mean ± SE	317.00 ± 5.01 ^a^	113.00 ± 4.06 ^e^	185.33 ± 2.76 ^d^	237.00 ± 4.66 ^c^	281.33 ± 2.13 ^b^
	LSD0.05 = 12.026					
	*t*-test	-	111.73 ***	−11.48 ***	−21.87 ***	−29.49 ***
GSH U/g	Mean ± SE	310.33 ± 2.56 ^a^	104.00 ± 3.84 ^e^	154.00 ± 2.55 ^d^	186.66 ± 2.37 ^c^	253.33 ± 4.26 ^b^
	LSD0.05 = 10.583					
	*t*-test	-	35.43 ***	−09.85 ***	−25.25 ***	−20.72 ***
TAC U/g	Mean ± SE	5.40 ± 0.10 ^a^	0.87 ± 0.02 ^e^	1.80 ± 0.04 ^d^	2.53 ± 0.04 ^c^	3.29 ± 0.05 ^b^
	LSD0.05 = 0.189					
	*t*-test	-	41.60 ***	−27.21 ***	−25.69 ***	−31.28 ***
SOD U/g	Mean ± SE	342.33 ± 2.07 ^a^	110.33 ± 1.52 ^e^	238.00 ± 2.28 ^d^	257.66 ± 2.78 ^c^	293.66 ± 3.31 ^b^
	LSD0.05 = 5.688					
	*t*-test	-	211.78 ***	−063.48 ***	−47.44 ***	−091.16 ***
CAT U/g	Mean ± SE	7.46 ± 0.09 ^a^	2.01 ± 0.02 ^e^	2.53 ± 0.04 ^d^	3.70 ± 0.07 ^c^	4.61 ± 0.09 ^b^
	LSD0.05 = 0.226					
	*t*-test	-	60.25 ***	−7.85 ***	−28.41 ***	−27.67 ***

*t*-test values, ***: means significant at *p* < 0.001. a, b, c, d, or e are ANOVA analysis significantly different at *p* < 0.05, LSD: least significant difference.

## Data Availability

The data presented in this study are available on request from the corresponding author.

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
