# Peer review of "Enhancement of the Protective Activity of Vanillic Acid against Tetrachloro-Carbon (CCl4) Hepatotoxicity in Male Rats by the Synthesis of Silver Nanoparticles (AgNPs)"

_molecules, 2022, doi:10.3390/molecules27238308_

Round 1

Reviewer 1 Report

you need to add the ethical number approval for experimental animals

remove the horizontal lines in all the figs

improve the English, there are a lot of typists error

give more detail about the homogenation

clarify the objective.

the characterization of the nanoparticles are not enough

Author Response

Reviewer 1:

Q1: you need to add the ethical number approval for experimental animals.

R1: The ethical number was added.

Q2: remove the horizontal lines in all the figs.

R2: Done

Q3: improve the English, there are a lot of typists error

R3: The MS was thoroughly revised and the editing certificate was provided.

Q4: give more detail about the homogenation

R4: More details about the homogenation were added.

Q5: clarify the objective.

R5: Done

Q6: the characterization of the nanoparticles are not enough

R6: More characterizations were added to the MS.

Thanks a lot for your interesting comments that helped us in improving our MS.

Reviewer 2 Report

The paper presents interesting research but some information is missing throughout and the paper needs editing for English grammar. Specific comments are below.

Lines 38-39: The sentence doesn't make sense. The end of the sentence should be "...by numerous biochemical processes that cause oxidative damage to proteins, DNA, and lipids."

Lines 45-46:  The end of the sentence should be "...used in cosmetics, fruit flavorings, cigarettes, pharmaceuticals, alcoholic drinks, flavoring agents, and fragrances."  What is polymer sectors in the context of this sentence?

Line 51:  What kind of activity does vanillic acid alleviate?

Lines 54-55:  If vanillic acid is antihypertensive, then shouldn't it decrease systolic and diastolic blood pressure?

Line 63:  MSH should be defined.

Lines 78-79:  Should this sentence be: "The biomedical uses of silver-based..."

Line 83:  Silymarin should be defined in the introduction so the reader can understand why it was used in the study. What is it? What does it do?

Section 2.3:  The methods for this section need more details.  What was the duration of the exposure to CCl4?  The abstract indicates 4 weeks but this needs to be mentioned in the methods.  Also, how often and for how long were the other chemicals administered? And when were they administered in relation to the CCl4 exposure? And by what route were they administered? 

Lines 130-131:  What does the "m" mean in "1m of 1% vanillic acid" and "1m of 1% np vanillic acid"?  Is it mL? Is it mg?

Figures 3, 4, and 5:  Why do the figures have the legend for the outcome parameter in only some parts of the figure? For example, figure 3c has the line with a circle in it at the bottom indicating how ALP is shown in the graph, and then there is a lighter colored line with a circle next to it that does not have anything written next to it. what is that? Why does figure 3c and figure 3d have this but not figure 3a and 3b? I think all of the lines with circles in them from all the figures should be deleted. Each figure has only one graph line in it for one outcome so they do not need to be labeled with a legend.

Line 208: According to Table 1, in G2, CCl4 toxicity decreased direct bilirubin, it did not increase direct bilirubin.

Lines 216-217:  According to Table 1, G3, G4, and G5 increased direct bilirubin, not decreased it.

Line 219:  It was already mentioned in section 3.1 that silymarin is a drug that protects the liver. It does not need to be repeated here or in section 3.3.

Author Response

Reviewer 2

Q1: The paper presents interesting research but some information is missing throughout and the paper needs editing for English grammar. Specific comments are below.

R1: English editing throughout the MS was done.

Q2: Lines 38-39: The sentence doesn't make sense. The end of the sentence should be "...by numerous biochemical processes that cause oxidative damage to proteins, DNA, and lipids."

R2: The sentence was corrected, thanks.

Q3: Lines 45-46:  The end of the sentence should be "...used in cosmetics, fruit flavorings, cigarettes, pharmaceuticals, alcoholic drinks, flavoring agents, and fragrances."  What is polymer sectors in the context of this sentence?

R3: Done, thanks.

Q4: Line 51:  What kind of activity does vanillic acid alleviate?

R4: It decreases neuroplasticity impairment and battling depression.

Q5: Lines 54-55:  If vanillic acid is antihypertensive, then shouldn't it decrease systolic and diastolic blood pressure?

R4:Yes, of course, we changed by to for and so the meaning became clear,  thanks.

Q5: Line 63:  MSH should be defined.

R5: It is melanocyte-stimulating hormone, we added it to the MS, thanks.

Q6: Lines 78-79:  Should this sentence be: "The biomedical uses of silver-based..."

R6: Yes, you are right, we changed it, thanks.

Q7: Line 83:  Silymarin should be defined in the introduction so the reader can understand why it was used in the study. What is it? What does it do?

R7: A short paragraph on silymarin was added to the MS, thanks.

Q8: Section 2.3:  The methods for this section need more details.  What was the duration of the exposure to CCl4?  The abstract indicates 4 weeks but this needs to be mentioned in the methods.  Also, how often and for how long were the other chemicals administered? And when were they administered in relation to the CCl4 exposure? And by what route were they administered? 

R8: CCl4 was administered at 8:00 am in the 1st and the 4th day every week for 4 weeks. Administeration rote was added in the MS, it was by intraperitoneal injection in the lower left or right quadrant of the abdomen. Treatment was daily for G3,G4 and G5. All inquiries was amended in the MS in red color., thanks.

Q9: Lines 130-131:  What does the "m" mean in "1m of 1% vanillic acid" and "1m of 1% np vanillic acid"?  Is it mL? Is it mg?

R9: It is 1mL., we corrected it, thanks.

Q10: Figures 3, 4, and 5:  Why do the figures have the legend for the outcome parameter in only some parts of the figure? For example, figure 3c has the line with a circle in it at the bottom indicating how ALP is shown in the graph, and then there is a lighter colored line with a circle next to it that does not have anything written next to it. what is that? Why does figure 3c and figure 3d have this but not figure 3a and 3b? I think all of the lines with circles in them from all the figures should be deleted. Each figure has only one graph line in it for one outcome so they do not need to be labeled with a legend.

R10: Fig3,4, and 5 were reloaded after correction, thanks.

Q11: Line 208: According to Table 1, in G2, CCl4 toxicity decreased direct bilirubin, it did not increase direct bilirubin.

R11: Of course, it must be increased in parallel with the total bilirubin, we revised the original results and found the right values and corrected them in the MS, thank you very much for your valuable note.

Q12: Lines 216-217:  According to Table 1, G3, G4, and G5 increased direct bilirubin, not decreased it.

R12: We corrected the result in Table 1, so, it is OK now.

Q13: Line 219:  It was already mentioned in section 3.1 that silymarin is a drug that protects the liver. It does not need to be repeated here or in section 3.3.

R13: We removed this sentence

Thank you very much for your valuable comments that helped us in improving our MS.

Reviewer 3 Report

The manuscript “Enhancement of the protective effect of vanillic acid against tet-2 rachloro-carbon (CCl4) hepatotoxicity in Male rats by synthesis 3 of silver nanoparticles (AgNPs)” is suitable for publication in the Molecules.

The authors investigated the synthesis and characterization of silver nanoparticles using vanillic acid and silymarin. Also, nanoparticles were tested to protect against CCl4 hepatotoxicity in male rats.

Specific comments:

  1. Abstract: 3rd group of mice was repeated two times, but there is no 4th group. Please, check where is the mistake. 
  2. The biological part of the Manuscript is precise and written and discusses better than the characterization of nanoparticle parts. The spectra of colloidal nanoparticles should be printed in some program (Origine, Excel) whit bold lines and ordinated. 
  3. When analyzing the TEM image, we can talk about the morphology of the particles. Now, we can say that particles are spherical. But we cannot generalize the size of the particles in the entire solution by analyzing only ten particles seen in the picture. According to the TEM picture, nanoparticles are between 10 and 15 nm in size. A more complete study of particle size distribution can be obtained by measuring it on a Zetasizer.
  4. In the conclusion part, there is no conclusion about the characterization of nanoparticles. 

The paper should be rewritten in the first part about the characterization of nanoparticles. I believe that it should be published after MAJOR revision. 

Author Response

Reviewer 3

The manuscript “Enhancement of the protective effect of vanillic acid against tet-2 rachloro-carbon (CCl4) hepatotoxicity in Male rats by synthesis 3 of silver nanoparticles (AgNPs)” is suitable for publication in the Molecules.

Thanks a lot!

Q1: Abstract: 3rd group of mice was repeated two times, but there is no 4th group. Please, check where is the mistake. 

R1: We corrected it, thank you very much for your valuable note.

Q2: The biological part of the Manuscript is precise and written and discusses better than the characterization of nanoparticle parts. The spectra of colloidal nanoparticles should be printed in some program (Origine, Excel) whit bold lines and ordinated. 

R2: we added more characterization, improved the figure of the spectra and added a new figure for calculating AgNPs size using dynamic light scattering (DLS) spectrum of the synthesized nanoparticles.

Q3: When analyzing the TEM image, we can talk about the morphology of the particles. Now, we can say that particles are spherical. But we cannot generalize the size of the particles in the entire solution by analyzing only ten particles seen in the picture. According to the TEM picture, nanoparticles are between 10 and 15 nm in size. A more complete study of particle size distribution can be obtained by measuring it on a Zetasizer.

R3: added a new figure for DLS spectrum of the synthesized nanoparticles.

Q4: In the conclusion part, there is no conclusion about the characterization of nanoparticles. 

 R4: We added some characters of the synthesized AgNPs in the conclusion (all changes are red colored).

Q5: The paper should be rewritten in the first part about the characterization of nanoparticles. I believe that it should be published after MAJOR revision. 

R5: We revised the MS carefully and added new information concerning the AgNPs.

Thank you very much for your interesting comments that helped us in improving our MS.

Looking forward to receiving your positive decision soon.

Round 2

Reviewer 1 Report

I do not have a program to check the plagiarism, so please do it in the journal system. It is essential not to exceed 25%.

Still, the characterizations are not enough

Author Response

Dear editor of Molecules,

It is my pleasure to resubmit the revised version of our MS entitled “Enhancement of the protective activity of vanillic acid against tetrachloro-carbon (CCl4) hepatotoxicity in Male rats by synthesis of silver nanoparticles (AgNPs) “ according to the reviewer’s comments. Below is a point-by-point response to every reviewer’s comment. Changes from the first revised version are given in red color. The MS was also thoroughly reviewed for any English or typing mistakes (a certificate was uploaded). Looking forward to receiving your positive decision soon

Regards

Response to review comments:

Reviewer 1:

Q1: I do not have a program to check the plagiarism, so please do it in the journal system. It is essential not to exceed 25%.

R1: The MS was tested for plagiarism, and the percentage was lower than 25 when we removed the addresses and references.

Q2: Still, the characterizations are not enough

R2: we increased the AgNPs characterization.

Thank you very much again for your comments that helped us in improving our MS.

Looking forward to receiving your positive decision soon.

Prof. Haddad El Rabey
